# NATURALPROOFS: Mathematical Theorem Proving in Natural Language

**Sean Welleck**[1,2]**, Jiacheng Liu**[1]**, Ronan Le Bras**[2]**,**
**Hannaneh Hajishirzi**[1,2]**, Yejin Choi**[1,2]**, Kyunghyun Cho**[3,4]
[1]Paul G. Allen School of Computer Science & Engineering, University of Washington
[2]Allen Institute for Artificial Intelligence
[3]New York University
[4]CIFAR Fellow in Learning in Machines & Brains
`wellecks@uw.edu`

## Abstract

Understanding and creating mathematics using natural mathematical language – the mixture of symbolic and natural language used by humans – is a challenging and important problem for driving progress in machine learning. As a step in this direction, we develop NATURALPROOFS, a multi-domain corpus of mathematical statements and their proofs, written in natural mathematical language. NATURAL-PROOFS unifies broad coverage, deep coverage, and low-resource mathematical sources, allowing for evaluating both in-distribution and zero-shot generalization. Using NATURALPROOFS, we benchmark strong neural methods on mathematical reference retrieval and generation tasks which test a system's ability to determine key results that appear in a proof. Large-scale sequence models show promise compared to classical information retrieval methods, yet their performance and out-of-domain generalization leave substantial room for improvement. NATURAL-PROOFS opens many avenues for research on challenging mathematical tasks.[1]

## 1 Introduction

Solving the problem of understanding and creating mathematics using *natural mathematical language* – the mixture of symbolic and natural language used by humans – is a path towards developing agents capable of reasoning. The mixture of symbolic and natural text, along with the existence of a formal counterpart, offers a unique setting for studying reasoning that complements research involving natural language alone or purely within a formal system. Constructing a mathematical proof involves symbolic manipulation, logical and analogical reasoning, as well as knowledge retrieval. Common sense and natural language abilities are needed to articulate the proof in a concise, comprehensible form. Moreover, systems that operate on mathematical text have applications in education and scientific discovery, while bridging informal and formal mathematics can be a key driver of progress in automated reasoning [5, 20, 37].

Recently, techniques from natural language processing have driven advances in *formalized mathematics* (e.g. Polu and Sutskever [30], Rabe et al. [31], Wu et al. [47]), in which mathematics is written in a verifiable formal language that resembles source code, such as Mizar [41], Lean [7], or Metamath [26]. However, this setting does not directly address the *informal* aspect of human mathematics, which is conveyed with a mixture of symbolic and natural language [13]. This aspect is crucial, since advancing *human understanding* is a goal of mathematics [40], and a significant fraction of mathematical knowledge is in natural language text [37].

---

[1]Dataset and code available at `https://github.com/wellecks/naturalproofs`.

35th Conference on Neural Information Processing Systems (NeurIPS 2021) Track on Datasets and Benchmarks.

| Source | ProofWiki |
|---|---|
| **Theorem** | **Category of Monoids is Category** |
| | Let $\mathrm{Mon}$ be the category of monoids. |
| | Then $\mathrm{Mon}$ is a metacategory. |
| **Proof** | Let us verify the axioms $(C1)$ up to $(C3)$ for a metacategory. We have |
| | Composite of Homomorphisms on Algebraic Structure is Homomorphism, verifying $(C1)$. |
| | We have monoid $(S, \circ)$. Now, $(C2)$ follows from |
| | Identity Mapping is Left Identity and Identity Mapping is Right Identity. |
| | Finally, $(C3)$ follows from Composition of Mappings is Associative. |
| | Hence $\mathrm{Mon}$ is a metacategory. |

| Source | Textbook: Real Analysis |
|---|---|
| **Theorem** | Suppose that $f$ is continuous on the closed interval $[a, b]$ and differentiable on the |
| | open interval $(a, b)$, and $f(a) = f(b)$. |
| | Then $f'(c) = 0$ for some $c$ in the open interval $(a, b)$. |
| **Proof** | Since $f$ is continuous on $[a, b]$, $f$ attains a maximum and a minimum value on $[a, b]$ (Theorem 2.2.9). If these two extreme values are the same, then $f$ is constant on $(a, b)$, so $f'(x) = 0$ for all $x$ in $(a, b)$. If the extreme values differ, then at least one must be attained at some point $c$ in the open interval $(a, b)$, and $f'(c) = 0$, by Theorem 2.3.7. |

Table 1: Example theorems and their proofs from NATURALPROOFS. Given a theorem, the mathematical retrieval task consists of retrieving the references (underlined) that occur in its proof. NATURALPROOFS contains data from ProofWiki, Stacks, and two textbooks; we show two sources here and two other sources in Table 12. See Figure 2 and Figure 3 for data format details.

In this paper, we describe NATURALPROOFS, a multi-domain corpus of mathematical statements and their proofs, written in natural mathematical language. NATURALPROOFS contains *broad-coverage* data from ProofWiki,[2] *deep-coverage* data from the Stacks project,[3] and *low-resource, real-world* data from mathematics textbooks. NATURALPROOFS unifies these sources in a common schema and is made publicly available as a resource to drive progress on tasks involving informal mathematics, complementing existing work in this direction (e.g. [11, 12, 43]).

Using NATURALPROOFS, we consider *mathematical reference retrieval*, an analogue of premise selection [1, 12]: given a mathematical claim, retrieve the set of references (theorems, lemmas, definitions) that occur in its proof. This task represents a crucial facet of mathematical reasoning, in which a mathematician determines the key results that appear in a proof. As a bridge towards generative tasks using NATURALPROOFS, we consider *mathematical reference generation*, which requires additionally recovering the order and number of references in each proof. Progress on either task could enable educational applications, such as providing a student with hints or guidance.

In addition to standard *in-distribution* evaluation, the multi-domain nature of NATURALPROOFS allows for evaluating *out-of-distribution*, zero-shot generalization. We design an evaluation protocol that tests a system's ability to retrieve references for *novel* theorems in each setting, and benchmark methods based on large-scale neural sequence models [8, 21], including a strong *joint retrieval* method that better refines the top of the ranked list, as well as an *autoregressive* variant for reference generation. The neural methods are effective for in-domain retrieval compared to classical techniques, yet out-of-distribution generalization, leveraging symbolic mathematical content, and fully recovering a proof's references remain as fundamental challenges. NATURALPROOFS opens many possibilities for developing and evaluating machine learning methods on challenging mathematical tasks.

## 2 Related Work

**Machine learning for mathematical theorem proving.** A large portion of work integrating machine learning with mathematical reasoning has focused on formalized mathematics. Early work by Urban [41] used machine learning for selecting relevant premises in the Mizar mathematical library that are passed to an automated theorem prover, which was later explored with deep neural networks [1]. Bansal et al. [3] developed the HOList benchmark based on the HOL Light theorem prover, while

---

[2] `https://proofwiki.org/`
[3] `https://stacks.math.columbia.edu/`

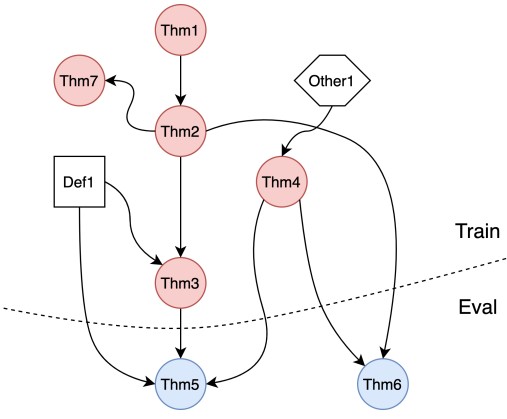

Table 2: The reference graph. Nodes are *statements* and edges are *reference* links. An edge pointing from A to B means that the proof for *theorem* B refers to *statement* A. Edges can start from any type of *statement*, but they always end at a *theorem*. In our tasks, the dataset is split so that all theorems in the evaluation sets are *leaf* nodes in the reference graph.

| | Source | All | PWiki | Stacks | RA | NT |
|---|---|---|---|---|---|---|
| **Theorem** | N | *32,579* | *19,734* | *12,479* | *298* | *68* |
| | Tokens | 46.7 | 38.2 | 60.6 | 33.6 | 23.7 |
| | Lines | 5.9 | 3.6 | 9.7 | 8.4 | 4.5 |
| | Refs | 1.8 | 2.8 | 0.2 | 0.0 | 0.0 |
| **Proof** | N | *32,012* | *19,234* | *12,479* | *235* | *64* |
| | Tokens | 181.5 | 199.3 | 155.5 | 128.9 | 97.2 |
| | Lines | 24.9 | 25.8 | 23.4 | 36.1 | 16.1 |
| | Refs | 5.6 | 7.4 | 3.0 | 1.6 | 0.9 |
| **Definition** | N | *14,230* | *12,420* | *1,687* | *86* | *37* |
| | Tokens | 48.4 | 45.0 | 73.2 | 58.6 | 32.6 |
| | Lines | 5.0 | 4.2 | 10.7 | 13.3 | 5.1 |
| | Refs | 2.9 | 3.3 | 0.4 | 0.0 | 0.0 |
| **Other** | N | *1,974* | *1,006* | *968* | – | – |
| | Tokens | 212.1 | 286.1 | 135.2 | – | – |
| | Lines | 34.4 | 46.7 | 21.7 | – | – |
| | Refs | 5.7 | 9.2 | 2.0 | – | – |

Table 3: NATURALPROOFS dataset statistics. Numbers represent mean value, except for "N" rows which represent count. **RA** is the Real Analysis textbook; **NT** is the Number Theory textbook. See Table 14 for detailed statistics.

others use the Coq [19, 48], Metamath [44, 42, 30], Isabelle [24], or Lean [14, 49] environments. These formalized settings differ from NATURALPROOFS, which uses mathematical language as humans write it. Szegedy [37] argues for leveraging both informal and formal mathematics through autoformalization. Wang et al. [43] explore translating between informal and formal mathematics, including via a dataset based on ProofWiki, though their dataset is not made available. Ferreira and Freitas [11, 12] propose a classification-based natural language premise selection task and a dataset based on ProofWiki, while NATURALPROOFS covers multiple domains and provides evaluation and benchmarks for full retrieval and generative tasks. The multiple informal domains, evaluation protocol, joint retrieval model, and reference generation task distinguish our work from previous work on ProofWiki and formalized mathematics.

**Mathematics and language benchmarks.** Several datasets evaluate a model's ability to solve multiple-choice algebraic word problems [35, 25, 2] or arithmetic problems [36] with varying degrees of natural language. Lample and Charton [22] evaluate neural sequence models on symbolic integration problems, while Hendrycks et al. [16] propose a benchmark based on math competition problems. NATURALPROOFS focuses on theorem proving rather than calculation, which we hypothesize evaluates different skills, and may prove useful in bridging formal and informal settings.

**Large-scale neural language models.** Large-scale unsupervised pretraining of language models has led to significant advances in many natural language processing domains (e.g. [8, 32, 33, 4]). Recent work suggests that these models store knowledge in their parameters [29], are capable of reasoning in mathematical [31, 47] and language [6, 38] domains, and are effective for information retrieval tasks [27, 28]. These advances motivate our work, which explores mathematical reasoning in natural language with large-scale language models through a retrieval task.

# 3 The NATURALPROOFS Dataset

The NATURALPROOFS Dataset is a large-scale, multi-domain dataset for studying mathematical reasoning in natural language. NATURALPROOFS consists of 32k theorem statements and proofs, 14k definitions, and 2k other types of pages (e.g. axioms, corollaries). Table 3 shows dataset statistics.

**Multi-domain.** NATURALPROOFS contains data derived from three domains:

1. *Broad-coverage* data that encompasses many mathematical topics (e.g. Set Theory, Analysis). We source data from ProofWiki (`https://proofwiki.org/`), an online compendium of mathematical proofs written by a community of contributors. Table 1 shows a theorem and its proof from ProofWiki. There are 31 top-level topic categories in this domain (§A.2).

2. *Deep-coverage* data that focuses on a single topic. We use the Stacks project (`https://stacks.math.columbia.edu/`), a collaborative web-based textbook of algebraic geometry written for graduate students and researchers. See Appendix Table 12 for an example.

3. *Low-resource, real-world* data that poses generalizability challenges to informal theorem proving systems. We use two open-source math textbooks with rich theorem-proof structures and reference links, specifically *Introduction to Real Analysis* (**RA** in short) by William F. Trench and *Elementary Number Theory: Primes, Congruences, and Secrets* (**NT** in short) by William Stein. See Table 1 and Appendix Table 12 for examples from each textbook.

NATURALPROOFS provides a common schema for mathematical statements, proofs, and the references that appear in each domain. Its multiple domains provide a challenging evaluation setting for models and opens opportunities for investigating domain transfer, out-of-distribution generalization, and methods for low-resource settings. This differs from existing resources that focus only on ProofWiki [11, 12], and reflects shifts in natural language processing towards multi-domain settings [45, 18], out-of-distribution generalization [23, 15, 39], and few- or zero-shot generalization in resource-constrained settings [4, 9].

**Structure.** Each *statement* in NATURALPROOFS is either a theorem or a definition. NATURAL-PROOFS provides the statement's title, contents, and references. The *contents* is a list of sequences, where each sequence contains one line of mixed text and LaTeX, with reference links displayed in their natural language forms. A *theorem* is associated with one or more proofs when available. A *proof* contains a title, contents, and references in the same format as a statement. Finally, we collect *other* pages (e.g. axioms, corollaries). A *reference* is a theorem, definition, or other page that is linked to within the contents of a statement or proof. Figure 3 shows the data format for theorems, definitions, and proofs in NATURALPROOFS. All statements and the reference links connecting them form a *reference graph*, shown in Table 2. The reference graph can contain cycles, e.g. `Pythagoras's Theorem` and `Sum of Squares of Sine and Cosine` refer to each other in their proofs.

**Data sources and preprocessing.** We describe how we retrieve data from each source and give an overview of preprocessing; for full details see Appendix A.1 and the Jupyter notebooks we release.

- **ProofWiki.** We download the public ProofWiki XML dump,[4] which contains a snapshot of all pages on ProofWiki. We filter pages according to manual rules (e.g. redirects, files, categories), and determine page type, title, contents, and references using each page's WikiMedia data structure.

- **Stacks.** We pull the Stacks GitHub repo,[5] which contains multiple LaTeX files for various sub-topics in algebraic geometry. We extract statements and proofs by LaTeX environment names. For example, the content enclosed by `\begin{theorem}` and `\end{theorem}` would be considered a theorem.

- **Textbooks.** We downloaded the LaTeX source of the **RA**[6] and **NT**[7] textbooks, and similarly extracted statements and proofs by environment names. In both textbooks, every statement is either a theorem or a definition – there are no statements that fall under "others".

## 4 NATURALPROOFS Reference Retrieval and Generation Tasks

NATURALPROOFS opens many possible machine learning tasks that involve natural mathematical language. We consider **mathematical reference retrieval**: given a theorem **x**, retrieve the set of references **y** that occur in its proof. An example is shown in Table 1, where the task is to retrieve the underlined references given the title and contents of the theorem `Category of Monoids is`

---

[4] `https://proofwiki.org/xmldump/latest.xml`. We use the November 12, 2020 version. ProofWiki is licensed under CC BY-SA 3.0.

[5] `https://github.com/stacks/stacks-project`. We use the April 15, 2021 version (commit 4df67b8). Stacks is licensed under GNU Free Documentation License.

[6] `https://digitalcommons.trinity.edu/mono/7/`. Retrieved on April 15, 2021. We did not use the supplementary materials. This textbook is licensed under CC BY-NC-SA 3.0.

[7] `https://github.com/williamstein/ent`. Retrieved on April 15, 2021. We provide a script to download and format the publicly available latex source.

| | Split | P+S | ProofWiki | Stacks | RA | NT |
|---|---|---|---|---|---|---|
| **Examples** $|\mathcal{E}|$ | total | **25,271** | **14,698** | **10,573** | **167** | **40** |
| | train | 21,446 | 12,424 | 9,022 | – | – |
| | valid | 1,914 | 1,139 | 775 | – | – |
| | test | 1,911 | 1,135 | 776 | 167 | 40 |
| **Refs** $|\mathcal{R}|$ | train | 42,056 | 28,473 | 13,583 | – | – |
| | valid | 45,805 | 30,671 | 15,134 | – | – |
| | test | 45,805 | 30,671 | 15,134 | 384 | 105 |
| **Refs/Ex** $|\mathbf{y}|$ | train | 5.9 | 7.5 | 3.6 | – | – |
| | valid | 5.6 | 7.5 | 2.9 | – | – |
| | test | 5.6 | 7.4 | 2.9 | 2.2 | 1.5 |

Table 4: NATURALPROOFS retrieval dataset statistics. **P+S** refers to the combined dataset from the ProofWiki and Stacks sources. **RA** (Real Analysis) and **NT** (Number Theory) are data from textbook sources that we use for zero-shot evaluation.

`Category`. As a proof is ultimately written as an ordered collection of statements with references often occurring more than once, we also consider **mathematical reference generation**: generate the *sequence* of references that occur in a given theorem's proof. These tasks represent a crucial aspect of theorem proving, in which a mathematician determines the key results that appear in a proof. Each task also enables educational applications, such as providing hints to a student about which previous results to use, or guidance on how to structure a proof.

**Reference retrieval and generation.** Each theorem $\mathbf{x}$ has a proof containing a sequence of references $\mathbf{y} = (\mathbf{r}_1, \ldots, \mathbf{r}_{|\mathbf{y}|})$, where each reference $\mathbf{r}_m \in \mathcal{R}$ is either a theorem, definition, or other statement (see §3). We consider two tasks: *retrieval* and *generation*.

In the *retrieval* task, given an input theorem $\mathbf{x}$, a model assigns a score to each reference in $\mathcal{R}$, inducing a ranked list $\hat{\mathbf{r}}^{(1)}, \ldots, \hat{\mathbf{r}}^{(|\mathcal{R}|)}$. These ranked references are evaluated against the ground-truth reference set using standard retrieval metrics such as mean average precision (MAP), recall (REC@$k$), and full recovery (FULL@$k$), which checks whether all references in the proof are in the top-$k$ predicted rankings. This reflects the goal of fully proving a theorem using a fixed number of results. For further details, refer to Appendix B.4.

In the *generation* task, a model produces a variable-length sequence of references $(\hat{\mathbf{r}}_1, \ldots, \hat{\mathbf{r}}_{|\hat{\mathbf{y}}|})$ given an input $\mathbf{x}$, with the goal of exactly matching the ground-truth reference sequence $(\mathbf{r}_1, \ldots, \mathbf{r}_{|\mathbf{y}|})$. Unlike retrieval, generation requires the model to correctly predict the total number of references, the number of occurrences of each unique reference, and their orders in the proof.

**Input-output examples.** Using NATURALPROOFS, we derive examples of the form $(\mathbf{x}, \mathbf{y})$, where $\mathbf{x} = (x_1, \ldots, x_T)$ is a theorem, and $\mathbf{y} = (\mathbf{r}_1, \ldots, \mathbf{r}_{|\mathbf{y}|})$ is the sequence of references that occur in the proof of $\mathbf{x}$. For retrieval, we transform each sequence into a set $\mathbf{y} = \{\mathbf{r}_1, \ldots, \mathbf{r}_{|\mathbf{y}|}\}$. The set of all references, $\mathcal{R}$, consists of theorems, definitions, and other statements (see §3). We use theorems with at least one proof that has at least one reference, resulting in a dataset with roughly 25k examples and a reference set $\mathcal{R}$ with 46k unique references. We partition the dataset into ProofWiki-only, Stacks-only, and textbook-only datasets. Table 4 summarizes the size, total references, and average references per example in each dataset.

**Training and evaluation splits.** We design training and evaluation splits that reflect the real-world scenario of proving *newly seen* theorems at evaluation time. This requires careful attention, since naively sampling evaluation examples would yield evaluation theorems that appear as references in the training set. To ensure that the theorems in the evaluation set have no overlap with the references in the training set, we form an evaluation set using a randomly sampled subset of *reference graph leaf nodes*, and use the remaining nodes as the training set (Table 2). We use roughly half of the evaluation set for validation and the other half for testing. Since evaluation theorems are not referred to in training examples, the reference set for training is smaller than that for evaluation (Table 4).

## 5 Methods

As benchmark methods for our tasks, we introduce two *parallel retrieval* methods, and a *sequential retrieval* method trained for sequence generation. See Appendix B for further implementation details.

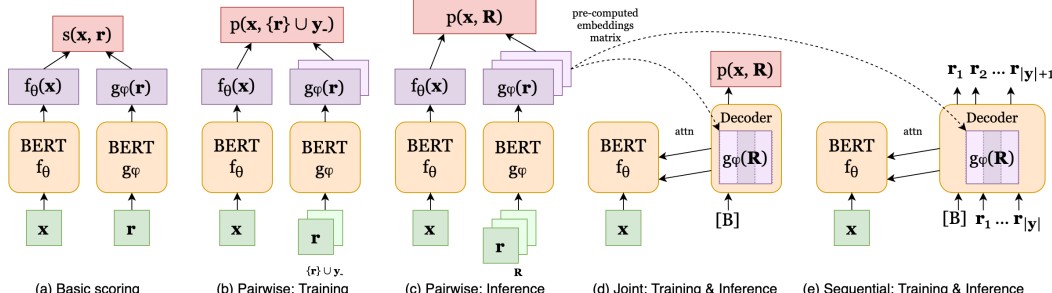

Figure 1: The pairwise, joint and sequential methods for mathematical reference retrieval.

**Parallel retrieval.** Given a theorem $\mathbf{x}$, a retrieval model should assign high scores to references in the proof of $\mathbf{x}$ and low scores to all other references, which corresponds to minimizing,

$$\mathcal{L}(\mathbf{x}, \mathbf{y}) = \mathrm{KL}\left(p_*(\mathcal{R}|\mathbf{x}) \| p_\theta(\mathcal{R}|\mathbf{x})\right) \tag{1}$$

$$\propto -\sum_{\mathbf{r} \in \mathbf{y}} \log \frac{\exp\left(s_\theta(\mathbf{x}, \mathbf{r})\right)}{\sum_{\mathbf{r}' \in \mathcal{R}} \exp\left(s_\theta(\mathbf{x}, \mathbf{r}')\right)} + \mathrm{const}, \tag{2}$$

where each distribution is over reference indices (i.e. in $\Delta^{(|\mathcal{R}|)}$), and $p_*(\mathbf{r}|\mathbf{x}) \propto \mathbb{I}[\mathbf{r} \in \mathbf{y}]$. The denominator requires scores $s_\theta(\mathbf{x}, \mathbf{r})$ for all $|\mathcal{R}|$ references, making backpropagation too expensive when a large-scale neural model is used to compute reference representations. As a result we consider two variants: a *pairwise* model that approximates Equation 1, and a *joint* model that computes Equation 1 but with implicit vector representations of each reference.

**Pairwise parameterization.** The pairwise model scores a reference $\mathbf{r}$ against a theorem $\mathbf{x}$ independent of other references, $s_\theta(\mathbf{x}, \mathbf{r}) = f_{\theta_1}^{\mathrm{thm}}(\mathbf{x})^\top g_{\theta_2}^{\mathrm{ref}}(\mathbf{r})$. The model is trained to contrast each positive reference with a set of negative references,

$$\mathcal{L}(\mathbf{x}, \mathbf{r}, \mathbf{y}_-) = -\log \frac{\exp(s_\theta(\mathbf{x}, \mathbf{r}))}{\exp(s_\theta(\mathbf{x}, \mathbf{r})) + \sum_{\mathbf{r}_- \in \mathbf{y}_-} \exp(s_\theta(\mathbf{x}, \mathbf{r}_-))}, \tag{3}$$

where $\mathbf{r}$ is a reference that occurs in the proof of $\mathbf{x}$, and $\mathbf{y}_-$ is a (small) set of negative references. At inference time, every reference is scored against an input theorem $\mathbf{x}$, inducing a ranking $\mathbf{r}^{(1)}, \ldots, \mathbf{r}^{(|\mathcal{R}|)}$. The scoring, training, and inference are illustrated in Figure 7 (a-c), respectively.

In practice, we use two instances of BERT [8] and in-batch negatives as in [21]. The pairwise model represents retrieval methods such as the dense passage retriever [21] and similar methods [27], and allows for evaluating large-scale sequence models on mathematical reference retrieval.

**Joint parameterization.** The joint model scores all references in a single pass,

$$p_\theta(\mathcal{R} \mid \mathbf{x}) = \mathrm{softmax}\left(\mathbf{R} f_\theta(\mathbf{x})\right), \tag{4}$$

where $\mathbf{R} \in \mathbb{R}^{|\mathcal{R}| \times d}$ is a reference embedding matrix and $f_\theta(\mathbf{x}) \in \mathbb{R}^d$ is a neural theorem encoder. This model has the advantage of computing the loss denominator in Equation 1 over *all* references rather than a subset of negatives. However, it must learn implicit representations of each reference without observing reference contents. To give the model access to representations that were learned using reference contents, we populate its embedding matrix as,

$$\mathbf{R} = \begin{bmatrix} — & g^{\mathrm{ref}}(\mathbf{r}_1) & — \\ & \cdots & \\ — & g^{\mathrm{ref}}(\mathbf{r}_{|\mathcal{R}|}) & — \end{bmatrix}, \tag{5}$$

where $g^{\mathrm{ref}}(\mathbf{x})$ is from a trained pairwise model. The joint model is illustrated in Figure 7 (d).

**Sequential generation and retrieval.** Finally, we consider an autoregressive model (Figure 7 (e)),

$$p_\theta(\mathbf{r}_1, \ldots, \mathbf{r}_{|\mathbf{y}|} \mid \mathbf{x}) = \prod_{t=1}^{|\mathbf{y}|+1} p_\theta(\mathbf{r}_t | \mathbf{r}_{<t}, \mathbf{x}), \tag{6}$$

|  |  | ProofWiki | | | | | Stacks | | | | |
| --- | --- | --- | --- | --- | --- | --- | --- | --- | --- | --- | --- |
|  |  | mAP | R@10 | R@100 | Full@10 | Full@100 | mAP | R@10 | R@100 | Full@10 | Full@100 |
| Random | | 0.04 | 0.00 | 0.19 | 0.00 | 0.00 | 0.07 | 0.05 | 0.60 | 0.00 | 0.13 |
| Frequency | | 3.38 | 5.90 | 24.30 | 0.44 | 2.29 | 0.91 | 1.76 | 11.27 | 0.13 | 2.45 |
| TF-IDF | | 6.19 | 10.27 | 23.09 | 4.14 | 9.43 | 13.64 | 25.46 | 47.36 | 18.94 | 37.76 |
| BERT (P+S) | +pair | 13.54 | 20.10 | 58.75 | 6.17 | 31.28 | 18.58 | 34.42 | 71.80 | 28.48 | 65.21 |
|  | +joint | 32.71 | 37.59 | 73.72 | 17.71 | 48.90 | 26.88 | 35.71 | 72.68 | 28.99 | 66.11 |
| BERT (P/S) | +pair | 16.82 | 23.73 | 63.75 | 7.31 | 38.50 | 20.93 | 37.43 | **74.21** | 30.03 | **66.37** |
|  | +joint | **36.75** | **42.45** | **75.90** | **20.35** | **50.22** | **28.32** | **39.10** | 73.61 | **31.96** | 65.59 |

Table 5: *In-domain* performance on the mathematical reference retrieval task (test set). **BERT (P/S)** is finetuned on the part of dataset with the same source as the evaluation set, whereas **BERT (P+S)** is finetuned on the combined dataset from ProofWiki and Stacks sources. Recall is micro-averaged.

where $\mathbf{r}_{|\mathbf{y}|+1}$ is a special $\langle\text{eos}\rangle$ token denoting the end of the reference sequence. The autoregressive model is trained to maximize the log-likelihood of ground-truth reference sequences. Unlike the parallel retrieval models, this model predicts the order and total number of references and can predict multiple occurrences of each reference. It also adjusts its predictions based on preceding predictions.

For generation, a standard decoding algorithm (e.g. beam search) is used to generate a reference sequence $\hat{\mathbf{y}} = (\hat{\mathbf{r}}_1, \ldots, \hat{\mathbf{r}}_{|\hat{\mathbf{y}}|} \langle\text{eos}\rangle)$. For retrieval, we populate a ranked list using generations $\{\hat{\mathbf{r}}_1, \ldots, \hat{\mathbf{r}}_{|\hat{\mathbf{y}}|}\}$ followed by references ordered according to the first step's probabilities, $p_\theta(\mathbf{r}_1|\mathbf{x})$.

# 6  Experiments

First, we benchmark the neural retrieval methods (§5) on mathematical reference retrieval in terms of their *in-domain* performance (Table 5) and their *out-of-domain* performance on an evaluation set formed from the textbooks in NATURALPROOFS (Table 7). We perform several analyses to better understand each method's strengths, weaknesses, and the factors that contribute to their performance.

**Experimental setup.** For the pairwise model, we use separate instances of `bert-base-cased` to parameterize the theorem encoder $f_{\theta_1}^{\text{thm}}$ and reference encoder $g_{\theta_2}^{\text{ref}}$. We implement the autoregressive model as a sequence-to-sequence encoder-decoder model. Following Rothe et al. [34], we parameterize the encoder and decoder using separate instances of `bert-base-cased`. This allows for initializing with pairwise model components. The joint retrieval model is implemented as a one-step variant of the autoregressive model. Pairwise models are trained for 500k steps (50 epochs for the autoregressive and joint models), evaluated every 5k steps (or 5 epochs) and the model with the highest validation mAP is selected for final evaluation. Refer to Appendix B for further details.

**Retrieval metrics.** We evaluate with standard retrieval metrics – mean average prevision (mAP) and recall@$k$ (R@$k$) – and a Full@$k$ metric measuring whether *all* true references are retrieved in the top-$k$ results. We use $k = 10$ and $k = 100$ for our evaluation. Refer to Appendix B.4 for definitions.

**In-domain performance.** The BERT-based retrieval models show strong in-domain performance compared to the classical TF-IDF and naive baselines in terms of average precision, recall, and the ability to fully recover all true references within the top-$k$ results, as seen in Table 5. On both ProofWiki and Stacks, the pairwise models outperform TF-IDF, with improvements that are consistent across reference types (Appendix Table 17).

Joint parameterization substantially improves over the pairwise models that are the starting point of joint training. On ProofWiki, the joint model ranks roughly 4 out of every 10 true references within its top 10 rankings (R@10 42.45) compared to 1 out of 10 for TF-IDF, and an impressive 75% within its top 100. For roughly half of the theorems, the joint model's top 100 references contain *all* of the references needed to prove the theorem (Full@100 50.22). On Stacks the recall@10 is similar at roughly 40%, with a higher full recovery rate of 66% for the top 100 results.

The gains from the joint parameterization are most prominent on ProofWiki, e.g. increasing mAP from 16.82 to 36.75. Joint parameterization particularly excels at refining the top of the ranked list compared to pairwise parameterization; the percentage improvement in the @10 metrics are larger than those for @100 metrics. On Stacks, the improvements are more modest: though mAP improves by 40%, the other metrics are relatively close, suggesting that advances beyond the joint

| Source | ProofWiki | | |
|---|---|---|---|
| **Theorem** | **Category of Monoids is Category** | | |
| | Let Mon be the category of monoids. | | |
| | Then Mon is a metacategory. | | |
| | **Ground-Truth Reference** | **Rank (Pairwise)** | **Rank (Joint)** |
| | Metacategory | 1 | 1 |
| | Identity Mapping is Left Identity | 4 | 5 |
| | Identity Mapping is Right Identity | 5 | 4 |
| | Monoid | 11 | 2 |
| | Composition of Mappings is Associative | 21 | 8 |
| | Identity Mapping is Automorphism | 117 | 64 |
| | Composite of Homomorphisms is Homomorphism | 261 | 54 |
| **Rank** | **Reference (Pairwise)** | **Reference (Joint)** | |
| 1 | *Metacategory* | *Metacategory* | |
| 2 | Monoid Category is Category | *Monoid* | |
| 3 | Monoid Category | Identity Morphism | |
| 4 | *Identity Mapping is Left Identity* | *Identity Mapping is Right Identity* | |
| 5 | *Identity Mapping is Right Identity* | *Identity Mapping is Left Identity* | |
| 6 | Category | Associative | |
| 7 | Composition of Morphisms | Identity (Abstract Algebra)/Two-Sided Identity | |
| 8 | Dual Category is Category | *Composition of Mappings is Associative* | |
| 9 | Identity Morphism | Composition of Morphisms | |
| 10 | Morphism Category | Semigroup | |

Table 6: Retrieval for a representative theorem. Top: predicted ranks for ground-truth references using the pairwise (left) and its joint (right) BERT models. Bottom: top 10 retrievals from the pairwise (left) and joint (right) models. A retrieved reference is italicized when it is a ground-truth reference.

| | Real Analysis | | | Number Theory | | |
|---|---|---|---|---|---|---|
| | **mAP** | **R@10** | **Full@10** | **mAP** | **R@10** | **Full@10** |
| TF-IDF | **15.79** | **34.65** | **27.54** | **16.42** | 39.62 | 30.00 |
| BERT-pair (P) | 13.24 | 24.01 | 19.16 | 15.12 | **41.51** | **35.00** |
| +joint | 11.24 | 20.97 | 16.77 | 15.85 | 41.51 | 35.00 |
| BERT-pair (S) | 11.56 | 21.28 | 14.97 | 12.58 | 26.42 | 20.00 |
| +joint | 7.04 | 11.55 | 9.58 | 14.88 | 26.42 | 20.00 |

Table 7: *Zero-shot* retrieval performance on out-of-domain textbooks.

model are needed. This demonstrates the importance of evaluating on multiple domains: each domain presents novel challenges for driving advances in modeling. Finally, the BERT models trained on both ProofWiki and Stacks (**BERT (P+S)**) show the possibility of training a single multi-domain model, albeit with lower per-domain performance than the models trained individually on each domain.

**Qualitative evaluation.** Table 6 shows model predictions for a representative theorem, `Category of Monoids is Category`. The pairwise model retrieves three out of seven true references within its top 50 results, while the joint model retrieves five out of seven. The top 10 results for both models are comprised of references that are related to category theory, which is the subject of the theorem. This illustrates the model's ability to retrieve *relevant* references, while highlighting its inability to always perform the fine-grained distinction between a relevant reference and one that occurs in the ground-truth proof(s). Arguably, such a system is still useful for providing hints to a user, so long as the user is confident that all of the true references are in a reasonably small set of results.

**Out-of-domain performance.** While strong in-domain performance drives applications in scenarios where training data is available, an ambitious goal is building a system with mathematical retrieval skills that automatically generalize to new resources. To evaluate the retrieval methods in this zero-shot, out-of-domain setting, we use each textbook from NATURALPROOFS as an evaluation set. This tests situations where the same theorem is expressed using different language (e.g. Table 13), generalization across data formats, and whether retrieval ability from in-domain training transfers.

Table 7 shows the results. The pairwise BERT model trained on ProofWiki underperforms TF-IDF on the Real Analysis textbook, and has comparable performance on the Number Theory textbook. Joint training did not improve out of domain performance, despite its favorable in-domain impact.

| | Model | Sequence | | | | | Multiset | | Set | | |
|---|---|---|---|---|---|---|---|---|---|---|---|
| | | EM | Edit($\downarrow$) | BLEU$_4$ | BLEU$_2$ | Len | EM | F1 | EM | F1 | BLEU$_1$ |
| **Stacks** | *-set | 51.74 | 35.70 | 9.75 | 47.73 | 0.97 | 89.03 | 97.04 | 100.0 | 100.0 | 94.09 |
| | *-multiset | 49.42 | 38.13 | 9.71 | 47.71 | 1.00 | 100.0 | 100.0 | 100.0 | 100.0 | 100.0 |
| | *-halfseq | 0.00 | 70.49 | 6.13 | 12.08 | 0.30 | 0.00 | 56.86 | 0.65 | 58.01 | 16.87 |
| | Joint | 0.00 | 98.81 | 0.00 | **3.42** | 2.82 | 0.00 | **19.24** | 0.00 | **19.65** | **15.15** |
| | Autoregressive | **3.87** | **90.65** | 0.00 | 2.59 | **0.97** | **4.00** | 13.14 | **4.90** | 15.04 | 10.06 |
| **ProofWiki** | *-set | 18.09 | 58.51 | 7.18 | 29.50 | 0.83 | 49.96 | 82.57 | 100.0 | 100.0 | 65.57 |
| | *-multiset | 19.23 | 58.09 | 16.68 | 52.89 | 1.00 | 100.0 | 100.0 | 100.0 | 100.0 | 100.0 |
| | *-halfseq | 0.00 | 58.84 | 25.88 | 29.17 | 0.41 | 0.00 | 63.33 | 4.21 | 70.26 | 30.55 |
| | Joint | 0.00 | 93.03 | 0.00 | 6.88 | 1.42 | 0.09 | 25.30 | 0.18 | **30.76** | 19.27 |
| | Autoregressive | **3.69** | **84.30** | **5.48** | **11.90** | **1.18** | **3.78** | **25.61** | **4.65** | 28.97 | **20.81** |

Table 8: In-domain *generation* results. We show the autoregressive model, a retrieval-only baseline using the top-5 predictions from the joint retrieval model, and oracle benchmarks for correctly predicting the first half of the sequence (*-halfseq*), the full multiset with randomized order (*-multiset*), and the full set with randomized order (*-set*). The best model-based method is in bold.

Training BERT on ProofWiki outperforms training on Stacks, showing that the training domain impacts out-of-domain generalization. ProofWiki's broad coverage of mathematics may help the model generalize better than the deep, single-topic coverage in Stacks.

The BERT models show some evidence of generalizing to out-of-domain mathematical sources, yet they do not show an advantage over traditional retrieval methods despite strong in-domain performance. This aligns with recent findings about neural retrieval models in various zero-shot settings [39]. An exciting research direction is using NATURALPROOFS to develop and evaluate methods which improve not only in-domain performance, but out-of-domain generalization.

## 6.1 Reference Generation

Next, we establish a benchmark for recovering the *sequence* of references occurring in the proof of each theorem via the reference generation task (§4).

**Metrics.** We evaluate predicted reference sequences against ground-truth sequences using order-aware **sequence** metrics, as well as unordered **multiset** and **set**-based metrics. Sequence metrics include exact match (**EM**), edit-distance (**Edit**), standard **BLEU**$_4$ score which uniformly weights 1-4 gram precision, **BLEU**$_2$ with only 1-2 gram precision, and average length ratio $\frac{\text{predicted}}{\text{true}}$ (**Len**). Unordered metrics include exact match, **F1**-score (corpus level), and 1-gram precision **BLEU**$_1$.

**Methods.** We use the autoregressive model to generate a reference sequence for each theorem using beam search. As a retrieval-only baseline, we form a sequence using the joint retrieval model's top-5 predictions, ordered by retrieval score. To judge performance and provide a benchmark for future work, we provide three oracle baselines: correctly predicting the first half of the sequence (*-halfseq*), the full multiset of references with random order (*-multiset*), and the set with random order (*-set*).

**Results.** Table 8 shows the in-domain generation results. The task is challenging, with the autoregressive model exactly matching the ground-truth sequence roughly 3% of the time. The autoregressive model improves over the retrieval-only baseline on order-aware metrics, aside from **BLEU**$_2$ on Stacks. It does length-prediction reasonably well, with length-ratios of 0.97 and 1.18, yet the multiset and set metrics indicate that the autoregressive model struggles to correctly predict the correct references, even after discarding order. The oracle baselines indicate substantial room for future improvement– for instance, predicting only half of each sequence correctly would move ProofWiki **BLEU**$_4$ from 5.48 to 25.88. Developing models along the full spectrum from set-based retrieval, to reference generation, to full proof generation is an exciting use-case for NATURALPROOFS.

## 6.2 Ablation Studies

**Initialization and autoregressive retrieval.** As shown in Table 9, the autoregressive model trained for sequence generation substantially improves over the pairwise retrieval model, yet underperforms

| Init | Model | mAP |
|---|---|---|
| – | Pairwise | 16.99 |
| – | Autoregressive | 17.77 |
| $f^{\text{thm}}$ | Autoregressive | 25.07 |
| $f^{\text{thm}}, \mathbf{R}$ | Autoregressive | **35.37** |
| – | Joint | 18.71 |
| $f^{\text{thm}}$ | Joint | 28.95 |
| $f^{\text{thm}}, \mathbf{R}$ | Joint | **37.51** |

Table 9: Initializing with pairwise components, and autoregressive retrieval (ProofWiki).

| Train | | Eval | |
|---|---|---|---|
| Lang. | NatProof | PW | Stacks |
| ✓ | ✗ | 0.14 | 0.30 |
| ✗ | ✓ | 0.04 | 0.86 |
| ✓ | ✓ | **16.99** | **21.21** |

Table 10: Language pretraining and NATURALPROOFS finetuning (pairwise retrieval, mAP).

| | Title | Content | PW | Stacks |
|---|---|---|---|---|
| **TF-IDF** | ✗ | ✓ | 4.97 | 12.34 |
| | ✓ | ✗ | **8.10** | 12.69 |
| | ✓ | ✓ | 6.33 | **13.45** |
| **BERT** | ✗ | ✓ | 16.19 | 19.12 |
| | ✓ | ✗ | **24.48** | 19.15 |
| | ✓ | ✓ | 16.99 | **21.21** |

Table 11: Excluding (✗) the title or content of theorems and references (pairwise retrieval, mAP).

the joint model, which is trained specifically for retrieval. Initializing the joint and autoregressive models using the pairwise model was necessary for achieving high performance; in particular, the reference information conveyed through the embedding matrix (Equation 5) was crucial.

**Language pretraining and NATURALPROOFS training.** The BERT model has two learning phases: pretraining on language data, and finetuning on NATURALPROOFS. As seen in Table 10, relying on language-pretraining alone without fine-tuning on NATURALPROOFS (top row) led to poor performance. Conversely, training from scratch on NATURALPROOFS (middle row) was unsuccessful, suggesting that language pretraining served as an effective initialization for mathematical retrieval.

**Title and content ablation.** Each theorem statement and reference consists of a title, as well as contents that is a mixture of symbolic mathematics and natural language. As seen in Table 11, ProofWiki's titles contain a large amount of useful information for retrieval– TF-IDF and the pairwise BERT model performed better with only access to titles. In principal, the title+content model could learn to ignore the contents if needed, so its lower performance shows a deficiency in the pairwise model. On Stacks, the model performs best with both sources of information, though the degree of improvement suggests that leveraging the mathematical content remains as a fundamental challenge.

# 7 Conclusion

Building agents that understand and create mathematics using *natural mathematical language* is a challenging research direction, providing a means for evaluating and developing machine learning methods capable of symbolic reasoning and natural language understanding. As a step in this direction, we develop NATURALPROOFS, a multi-domain dataset for studying mathematical reasoning in natural language. NATURALPROOFS allows for evaluating *in-domain* performance, and *out-of-domain* generalization in broad and deep coverage mathematics, as well as real-world, low-resource settings. We establish benchmarks for retrieval and generation tasks that represent key steps in real-world theorem proving, and are tractable, yet challenging, for current large-scale neural sequence models. NATURALPROOFS opens many promising avenues for future research.

**Broader Impacts Statement** Our work pertains to use of the natural language in mathematical theorem proving, and more generally reasoning in artificial intelligence. Although a general reasoning agent may present negative societal impacts, we do not foresee any immediate negative societal impact from the domain, dataset, tasks, and study that we present here. Instead, we foresee positive societal impacts through applications in education and scientific discovery that are enabled by systems which understand and create natural mathematical content.

**Acknowledgements** This work was funded in part by the Natural Sciences and Engineering Research Council of Canada (NSERC) (funding reference number 401233309), DARPA MCS program through NIWC Pacific (N66001-19-2-4031), and the Allen Institute for AI. We also thank Google Cloud Compute, as well as OpenAI. The authors would like to thank Deborah Ferreira, Jesse Han, Christian Szegedy, and Josef Urban for helpful discussions and feedback on earlier drafts.

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
