# OpenReview forum: "NaturalProofs: Mathematical Theorem Proving in Natural Language"
_NeurIPS.cc/2021/Track/Datasets_and_Benchmarks/Round1 — NeurIPS 2021 Datasets and Benchmarks Track (Round 1)_

### Official Review · Reviewer_28mF · 2021-07-03
**A dataset to select definitions that are relevant to prove a mathematical claim.**

**Rating:** 8
**Confidence:** 2
**Correctness:** Yes. The paper is soundless.
**Clarity:** 1. This is a dense paper, and that ma…

**Strengths:**

1. The work is very complete: it presents a dataset, a benchmark and also models for two different tasks. The number of experiments is very extensive.

2. The task is interesting and challenging, and although I have some doubts about the true utility of the task as it is proposed (see Weakness), the dataset seems to have been created carefully and in a curated way.

3. As I mention in the Correctness, the paper is soundless and the proposed methodology and benchmarks seems correct, IMO.


**Weaknesses:**

1.a I have some concerns on the use the authors make about the word ‘reasoning’, how they introduce the concept of ‘mathematical proving’, and what the dataset is intended for. Specially, for the case of the retrieval task, I’m not convinced whether this can be considered as a reasoning problem (such as for instance problems where multi-hop architectures are used) or on the contrary this is simply about finding references whose (BERT) representation will be close to the target claim (BERT) representation.

1.b Also, if I’m understanding correctly, at least for the retrieval task (and the models), the set of references R is known and fixed at inference time. If so, how is this connected to concepts such as ‘reasoning’ and ‘generalization’?

Could you share your thoughts on these?


2. As I mention in detail in the Clarity section, I found the paper a bit hard to follow.


**Additional Feedback:**

1. Lines 34-35/80-86 discuss about different sources of the dataset being broad-coverage, deep-coverage and low-resource, however I would appreciate if more context could be given for these terms (specially w.r.t to what it is considered low-resource in the context of mathematical proving). This is a paper describing a dataset and submitted to a Datasets and Benchmark track, and so it’s content should be presented in a very detailed and specific way.

2. Lines 87-88 mention that the corpus is multi-domain, however little is said about the specific domains we can find in the dataset unless the Appendix is read. I feel it’s necessary for a piece of work introducing a dataset. I feel this is also relevant to have a better understanding of the in-domain versus the out-of-domain experiments.

3. The pairwise approach uses a small set of negative references as a tractable alternative to the parallel strategy, could you expand on how that small set of negative references is selected?

4. For the mathematical reference generation, what guarantees do we have that the reference order from a given claim is unique?

5. I felt some parts of the paper include some over-claims. For instance:
Lines 126-127, would it be possible to soft them?
Lines 19-20, at this point, it gives the impression the paper will be doing mathematical proving, which is not truly the case.

6. Are the many references that are unique or not-frequent? If so, I wonder how the performance varies depending on their frequency?

7. I might be missing something, but, why is the average length ratio (line 242) an order-aware metric?


**Documentation:**

The code is uploaded and available on github and it also seems to be well-documented (both to download the data and to reproduce  the experiments).

**Ethics:**

I don’t see major ethical concerns with this paper

**Relation To Prior Work:**

1. The novelty of the work is not perfectly clear to me at the moment. The authors clarify this is an early work on mathematical proving that combines symbolic and natural language text content, but it is not clear whether this is also one of the first works on mathematical reference retrieval and generation, and I felt the current Related Work section couldn’t clarify as it should. Would it be possible to rephrase this bit in a future version of the paper?


**Summary And Contributions:**

This paper presents a dataset and a benchmark for the domain of mixed symbolic and natural language mathematical proving, namely the NATURALProofs dataset. The samples from the dataset come from different sources and more specifically ProofWiki, the Stacks project and a few mathematical textbooks available in LaTeX and that can be parsed to create a dataset for the purpose at hand. Additionally, the authors present baselines and evaluation metrics for two tasks:

(i) mathematical reference retrieval: given a mathematical claim (theorem + proof) the goal is identifying the set of *references*, i.e. theorems, lemmas, definitions, that are relevant in the proof.

(ii) mathematical reference generation: given a mathematical claim (theorem + proof) the goal is to generate an ordered sequence of the references used for the proof. This model can be also used as a retrieval model (although as the experiments show (and as expected) it will perform poorer than the models specifically designed for retrieval).

---

> ### Author Response · Authors · 2021-07-09
> **Response**
>
> Thank you for the positive and detailed review, and we appreciate the points about interesting and challenging tasks, extensive experiments, and correctness. Our responses to your feedback and questions are below:
>
> Re: “the word ‘reasoning’, how they introduce the concept of ‘mathematical proving’”
> - Thanks for the thought-provoking points: indeed defining reasoning itself is difficult. The retrieval task uses the intuition that when someone is presented with a theorem to prove, part of the reasoning process involves figuring out which known results will be used in the proof. It’s a great point that there are other aspects of the theorem proving process and the reasoning involved in it. By handling data collection, formatting, etc., we hope that NaturalProofs lowers the barrier for others to study further aspects of reasoning and theorem proving, in addition to making further progress on those we considered here.
>
> Re: “set of references R is known and fixed”
> - Just to clarify, R is not fixed: for instance, in the out-of-domain evaluation (Table 7) we evaluate on references from an unseen reference set.
>
> Thank you as well for the detailed additional feedback. Given the additional page for the final revision, we will incorporate these points along with further clarifying the related work.
>
> Re: definitions of broad/deep/low-resource
> - Thanks for pointing this out, we will include further details about the data in the main text versus the appendix, as well as definitions. To clarify:
>     - By broad-coverage, we mean that ProofWiki contains many subjects, (e.g. Set Theory, Analysis, see Figure 4).
>     - By deep-coverage, we mean that Stacks focuses on a single subject (algebraic stacks), and is written for graduate students and researchers in algebraic geometry (https://stacks.math.columbia.edu/about).
>     - The textbooks are low-resource in the sense that they don’t contain a substantial amount of examples which raises challenges due to not having any (or a large amount of) training data. An analogy is a low-resource language in translation.
>
> Re: negatives: We use in-batch negatives (Appendix B.1).
>
> Re: frequency and performance:
> - Looking into further questions like this is a great starting point for using the data and models!
>
> Re: unique reference order and length metric
> - Indeed there is no guarantee of a unique reference order- an analogous issue arises in translation, where word orders often lead to multiple plausible translations. To address this we report both order-aware and unordered metrics, both of which show large room for future improvement.
>
> Thanks again for the detailed feedback and review!

---

### Official Review · Reviewer_k6pE · 2021-07-05
**Feedback on NaturalsProof - Mathematical Theorem Proving in Natural Language**

**Rating:** 7
**Confidence:** 4
**Correctness:** The submission fits the theme.
**Clarity:** Clear and concise

**Strengths:**


Indeed, creating and formalizing mathematical problems using natural mathematical language itself is challenging.

I believe the paper has made a new contribution, given the following points:
•	Well written, clear and nicely structured. It is self-contained with background to the study and methods used.
•	The notations used are consistent.
•	The ablation to the study is clearly exposed.


**Weaknesses:**

The only demerits would be to include full set of assumptions of theoretical results which was not shown in the paper. Although, the checklist indicated that.

**Additional Feedback:**

None.

**Documentation:**

There's sufficient detail about the methods used.

**Ethics:**

No ethical concerns

**Relation To Prior Work:**

Clearly mentioned

**Summary And Contributions:**

The study presented an interesting corpus called NATURALPROOFS written in natural mathematical language, which is vital in multi-application domains, for example in education. This work provides a new direction in abstracting complex mathematical problems and challenging tasks

Indeed, creating and formalizing mathematical problems using natural mathematical language itself is challenging.

I believe the paper has made a new contribution, given the following points:
•	Well written, clear and nicely structured. It is self-contained with background to the study and methods used.
•	The notations used are consistent.
•	The ablation to the study is clearly exposed.

---

> ### Author Response · Authors · 2021-07-09
> **Response**
>
> Thank you for the positive review highlighting the new directions, presentation and structure, and experiments.
> Regarding the checklist, we put “[N/A] We did not include theoretical results” which we thought reflects the content of the paper, though please let us know if we misinterpreted this point.

---

### Official Review · Reviewer_7g61 · 2021-07-06
**NATURALPROOFS: Mathematical Theorem Proving in Natural Language**

**Rating:** 8
**Confidence:** 4
**Correctness:** The experiment was properly documented.

**Strengths:**

The dataset is a large-scale, multi-domain dataset for studying mathematical reasoning in natural language, consisting of 32k theorem statements and proofs, 14k definitions and 2k other types of pages—the multiple domains open opportunities for investigating domain transfer and methods for low-resource settings. The use of open-source maths textbooks with theorems and proofs is interesting.

**Weaknesses:**

The author should have not clearly explained the theoretical background of the work. Some crucial points like training the BERT model with the title and content were missing in the paper; the authors should have explained why they did not attempt this.

**Additional Feedback:**

The author introduced a work that can have a positive impact on mathematics education. More effort should be made to discuss this or other potential social impacts of NaturalProof.

**Clarity:**

The paper is clearly written. However, more details should be given in the methodology section; this will enhance reproducibility.

**Documentation:**

The URL provided provides sufficient details on the NaturalProofs Dataset, tokenized task data for mathematical reference retrieval and generation, preprocessing NaturalProofs and the task data, training and evaluation for mathematical reference retrieval and generation, pre-trained models for mathematical reference retrieval and generation.

**Relation To Prior Work:**

Some effort was made with regard to related work.

**Summary And Contributions:**

The authors developed NATURALPROOFS, a multi-domain corpus of mathematical statements and their proofs, written in natural mathematical language. Using the developed model, the authors benchmarked strong neural methods on mathematical reference retrieval and generation tasks which test a system’s ability to determine key results that appear in proof. NATURALPROOFS allows for evaluating in-domain performance and out-of-domain generalization in broad and deep coverage mathematics, as well as real-world, low-resource settings.

---

> ### Author Response · Authors · 2021-07-09
> **Response**
>
> Thank you for the positive and thoughtful review. Our responses to your suggestions are below:
>
> Re: “More details in the methodology section”
> - Thank you for the suggestion and feedback. With the extra page allowed in the final revision we’ll integrate more key details into the main text versus the Appendix. To ensure reproducibility we include full details in Appendix B along with code, but it’s a great idea to include more in the main text.
>
> Re: “training the BERT model with the title and content”
> - Please refer to Appendix B (“Model input format”) for details on how we trained BERT with the title and content.
>
> Re: “Positive impact on mathematics education”
> - We fully agree about the positive impacts on education! In the final revision we will comment further on this in the introduction and when discussing the tasks (for instance, reference retrieval could provide hints to a student).

---

### Decision · Program_Chairs · 2021-07-26

**Decision:**

Accept

**Comment:**

Reviewers univocally argue for acceptance of this paper that provides a carefully crafted, large-scale, multi-domain dataset for studying mathematical reasoning in NLP. Reviewers are also impressed, extensive number of experiments in this challenging domain. A few minor points raised by reviewers are responded to by the authors with proposed edits.